# Improving Phosphate Acquisition from Soil via Higher Plants While Approaching Peak Phosphorus Worldwide: A Critical Review of Current Concepts and Misconceptions [note 1]

**DOI:** 10.3390/plants13243478

**Published:** 2024-12-12

**Authors:** Jörg Gerke

**Affiliations:** Institut für Angewandte Wissenschaft, Ausbau 5, 18258 Rukieten, Germany; gerke.rukieten@t-online.de

**Keywords:** phosphate solubilization, carboxylates, soil P forms, chemical reactions of P mobilization, P sorption/desorption, phytate desorption, humic Fe/Al-P complexes, P uptake of mobilized P

## Abstract

Phosphate (P) is the plant macronutrient with, by far, the lowest solubility in soil. In soils with low P availability, the soil solution concentrations are low, often below 2 [µmol P/L]. Under these conditions, the diffusive P flux, the dominant P transport mechanism to plant roots, is severely restricted. Phosphate is sorbed into various soil solids, Fe/Al oxides, clay minerals and, sometimes overlooked, humic Fe/Al surfaces. The immobilization of P in soil is often the result of the diffusion of P into the internal surfaces of oxides or humic substances. This slow reaction between soil and P further reduces the availability of P in soil, leading to P fixation. The solubilization of soil P by root-released carboxylates is a promising way to increase the acquisition and uptake of P from P-fixing soils. Citrate and, sometimes, oxalate are effective with respect to additional P solubilization or P mobilization, which may help increase the diffusive P flux into the roots by increasing the P solution concentrations in the rhizosphere. The mobilization of humic-associated P by carboxylates may be an effective way to improve soil P solubility. Not only orthophosphate anions are mobilized by root-released carboxylates, but also higher phosphorylated inositol phosphates, as the main part of P esters in soil are mobilized by carboxylates. Because of the rather strong bonding of higher phosphorylated inositol phosphates to the soil solid phase, the mobilization step by carboxylates appears to be essential for plants to acquire inositol-P. The ecological relevance of P mobilization by carboxylates and its effect on the uptake of P by crops and grassland species are, at best, partially understood. Plant species which form cluster roots such as *white lupin* (*Lupinus albus* L.) or *yellow lupin* (*Lupinus luteus* L.) release high rates of carboxylates, mainly citrate from these root clusters. These plant species acquire fixed or low available P which is accessible to plants at rates which do not satisfy their P demand without P mobilization. And *white lupin* and *yellow lupin* make soil P available to other plants in mixed cropping systems or for subsequent plant species in crop rotations. The mobilization of P by carboxylates is probably also important for legume/grass mixtures for forage production. Species such as alfalfa, red clover or white clover release carboxylates. The extent of P mobilization and P uptake from mobilized P by legume/grass mixtures deserves further research. In particular, which plant species mostly benefit from P mobilization by legume-released carboxylates is unknown. Organic farming systems require such legume/grass mixtures for the introduction of nitrogen (N) by forage legumes into their farming system. For this agricultural system, the mobilization of soil P by carboxylates and its impact on P uptake of the mixtures are an important research task.

## 1. Introduction

Phosphorus, mainly present as orthophosphate (P) or its esters in the soil–plant system, is an essential macronutrient for higher plants. The global reserves are limited and soil P deficiency is widely distributed worldwide among agricultural soils [1,2], with parts of Africa, South America, Arabia and India being strongly P-impoverished [3]. On the other hand, parts of agricultural soils in Europe and China exhibit high accumulations of available P and even P leaching [3].

The limited P resources worldwide, strong P deficiency in soils around the globe and inefficient P acquisition of fertilizer P by plants [4] explain the need to improve the rate of transfer of P from soil to cultivated plants.

Without any doubt, the strategy to fertilize or overfertilize soils with mineral and organic P fertilizers cannot be maintained in the future. This P fertilization practice has led to an inequal P distribution worldwide considering the P availability in soils.

The problem cannot be solved alone by applying a rational P fertilization regime worldwide because inequality in soil P availability is partly linked to the global food trade system which becomes increasingly important.

For example, soyabeans are an important cash crop for farmers in South America, e. g., Brazil and Argentina, which export beans mainly to the European Union, where they are the main protein source for milk, egg and meat production. Phosphate in soyabeans is transported from low-P soils in South America to regions like Netherlands and Northwestern Germany, where P from liquid manure application arising from soyabean feed cattle or pigs is applied to even soils already highly provided with P. Phosphate-depleted soils in South America are further depleted by the export of soyabeans to regions with already high P levels which are increasingly overfertilized by soyabean P.

Considering the limited P reserves, nutrient recycling by local food chains is one answer to manage rational handling of the global restricted phosphate reserves.

This, however, is a political task which requires the support of local food production and consumption cycles in order to avoid the further accumulation of plant nutrients, mainly of P in richer countries.

It would also make sense to selectively apply mineral P fertilizers to soils low in available P. This, however, does not happen worldwide. Farmers in Africa are especially often not able to buy P fertilizers because of the lack of capital, and fertilizer transport and dealer costs [5]. As shown for Angola by Cordell and White [5], extreme high dealer costs indicate that corrupt political and economic structures affect the application of P fertilizers in the country. The establishment of local nutrient chains and the fight against political corruption are goals which should be approached. However, these political tasks will not be considered further here.

Similar to peak oil, peak phosphorus has been defined as the year where maximum P reserves are extracted. Cordell and White [5] compared several models which calculated the year of peak phosphorus to be between 2011 and 2084. Two other calculations which speculate about no increasing yearly phosphate rock consumption, worldwide, assume a constant mining for at least the next two hundred years. Despite the differences in the calculated peak *p*-values, it is clear that rock phosphates are a strongly limited resource compared to the yearly demand worldwide and that further P mining will aim to use reserves which are increasingly lower in P content. And it will lead to increasing fertilizer costs. About 90% of rock phosphate is used for food production, mainly P fertilizers [5]. A world population growing to 9 or 10 billion will demand increased food production and probably increased P fertilizer application mainly to soils low in available P. But increasing mineral P fertilizer applications will not be sufficient to improve P acquisition by agricultural plants, and increase agricultural plant yields. Additionally, farming strategies have to be applied to increase the production of food [6]. This is no argument to omit mineral P fertilizer application, but an argument to improve P acquisition by higher plants from soil at a given amount of available P.

The following chapters focus on soil P availability and the factors which affect it, and explain P acquisition by plants. The availability of P is a soil characteristic and P acquisition depends on the characteristics of the plants and its roots. The interaction of P availability and P acquisition parameters determines the quantity of P taken up by plants, as has been shown in detail by Tinker and Nye [7], and Jungk [8].

The main hypothesis is that under low soil P conditions, the release of efficient carboxylates by the roots is a promising and often essential way to improve the acquisition of soil P.

## 2. Soil Phosphate

Among the plant macronutrients which are mainly taken up by the roots, P has, by far, the lowest soil solubility, i.e., the P soil solution concentration in equilibrium with its solid phase is comparatively low. Nutrients are transported in the soil solution to roots via mass flow and diffusion before they are taken up by the roots [7,8]. As a rule, if the soil nutrient solution concentrations are high in relation to the plant’s demand, then mass flow is the dominant transport mechanism, but if the soil solution concentration is low, then diffusive flux is dominant. For P, diffusive flux to the roots is the dominant transport mechanism in soils, with mass flow accounting for less than 1–3% of P transport to roots [7,8].

The P soil solution concentration and P buffering, a measure for the desorption of P from the soil solid phase into the soil solution which is mathematically expressed as the slope of the P desorption curve, are the central parameters of soil P availability [9]. Both parameters, soil P solubility and P buffering, can be experimentally assessed by various methods.

Phosphate in the soil solution is usually determined after collecting by displacing, centrifugation or suction caps.

Available P at the soil solid phase is determined by several extracting solutions, e.g., with mineral or organic acids or a NaHCO_3_ solution. Also, soil extraction with anion exchangers is used to extract a measure of available soil P. The extracted P quantities of the routine procedures are related to the results of field trials by evaluating the effect of P fertilization according to the amount of P extracted by the respective routine method.

The third important P soil parameter is the quantity of total P in soil which is usually determined after ignition and/or applying harsh extracting solutions aiming to account for all of the soil P. Total P (P_t_) is no measure for available P. Total soil P is often in the range of 200–2000 [mg P/kg soil]. Available P, as the soil P part that is in exchange with soil solution P, accounts for between less than 1% and up to 15% of total P. Soil P solution concentrations are in the range of 1–10 [µM P]; in P-fixing soils, they are much lower, and in highly and overfertilized soils much higher, at up to 100 [µM P]. In most soils, far less than 0.1 per mile of P_t_ is actually in the soil solution.

These experimental approaches to soil P availability, however, do not explain the physico-chemical, chemical, biochemical and biological ways of P reactions in soil, which are the underlying mechanisms of P availability, acquisition and uptake.

It has been proposed that most soil P is in discrete fractions. These fractions can be sequentially extracted, leading to fractionation procedures, e.g., by Chang and Jackson [10], Kurmis [11], or Zhang and Kovar [12]. It has been assumed that these discrete fractions consist of Al or Fe phosphates under acidic conditions, and Ca phosphates under basic conditions in soil. Therefore, Al/Fe-associated P is extracted with alkaline solutions (NaOH) and Ca-associated P is extracted with acid solutions (HCl or H_2_SO_4_) via fractionation procedures.

Under the assumption of chemical equilibria in soil, the distribution of discrete P mineral fractions under varying chemical conditions can be calculated, as demonstrated in detail by Lindsay [13].

This view has been questioned. The results on soil P can be better explained by models of continuous P binding strength distribution in soils. Barrow [14] summarized some of these results. Phosphate fractionation procedures fail because discrete fractions are arbitrarily defined by the sequential fractionation procedures but do not achieve appropriate differentiation between Ca-P, Al-P and Fe-P compounds [11,15], and between inorganic and organic P [16].

By summarizing his work, Barrow [14] described that initially, after P application, the exchange between soil solution P and the soil solid phase adsorbing sites is dominant, followed by solid-state diffusion of the adsorbate into the adsorbent as shown by Strauss et al. [17] in the case of the diffusion of P into goethite. Similar reactions between goethite and heavy metals were found by the same scientific group. The reaction model of Barrow [14] explains three modes of reaction occurring in many soils after the application of water-soluble fertilizer P.

First, the Barrow model explains the adsorption/desorption hysteresis for P in soil, as principally shown in Figure 1.

And it explains the continuous decrease in soil solution P concentrations (P_l_) after the addition of water-soluble P fertilizer over a long period through the slow reaction between P and soil [18], which is principally shown in Figure 2.

For soil P availability, the desorption, and not the adsorption, curve behavior is important considering the plant root as a P sink which first depletes soil solution P, which, in turn, induces P desorption from the soil solid phase.

A third conclusion may also be drawn from the adsorption/diffusion model of Barrow that no P is principally inaccessible to the plant roots, but with increasing soil P depletion by the roots, the availability of the remaining P continuously decreases.

The desorption/diffusion model for the reaction of P with soil components is well founded by experimental results [17,19,20,21]. But it may be too restrictive because it only accounts for a part of the P sorbing sites in soil. Gerard [22] reviewed the relevance of clay minerals for the adsorption of P in soil and compared it to that of Fe/Al oxides in soil by evaluating P adsorption studies on soils from the last 70 years. Gerard [22] concluded that the P binding capacity of clay minerals is in the same range as that of Fe/Al oxides. He further concluded that at lower P concentrations, clay minerals may control P sorption, whereas at high P concentrations, oxides and P diffusion into amorphous regions of the oxides may control P sorption.

Oxides and clay minerals themselves only cover a part of the P binding sites in soils. Humic substances as P sorbents must be taken into account. The orthophosphate anion is not directly bound to humic molecules, but is bound via Fe (III) or Al (III) bridges to the humic phase [23,24,25]. Possibly, Cu (II) may also act as a bridging cation [26], whereas Ca (II) as a bridging cation may be of minor importance as evaluated from P adsorption studies on humic Fe (III) complexes with or without a 0.01 M CaCl_2_ matrix [27]. This does not mean that the production of humic P fertilizers using Ca^2+^ as a bridging cation, as suggested by Erro et al. [28], is useless since the concentration relations are different during fertilizer fabrication compared to the soil solid and soil solution chemistry. The relatively easy release of P from humic Ca-P complexes may represent an advantage for humic-based P fertilizer fabrication.

One main difference between oxide associated and humic Fe(Al)-associated P is that humic molecules with bound P can be dissolved in the soil solution, and may themselves diffuse into the roots.

Humic-associated P is often ignored as the main part of soil P. The main reason is due to the fact that the determination of humic-associated P is not achieved by often-applied conventional procedures.

The determination of organic P (P_o_) in soil and the soil solution often includes the determination of total P (P_t_) and that of inorganic P (P_i_) by photometric procedures such as the Murphy and Riley [29] procedure or similar photometric procedures, e.g., Ohno and Zibilske [30]. The difference of P_t_ and P_i_ is assumed to be organic P. During photometric analysis, the solution is acidified to pH values below 1. Under these reaction conditions, humic-associated P is immediately hydrolyzed from the organic phase and determined as free, inorganic P, e.g., desorbed from the Fe oxide, ignoring its real chemical status (see review in [16]). As an approximate rule, about 75–85% of humic-associated P_i_ can be hydrolyzed during photometric P_i_ determination; between 15 and 25% will be determined as organic phosphate. This means that via the difference method, a significant part of humic-associated P will be determined as phosphate esters. He et al. [31] used phosphatases and UV irradiation for the release of different forms of humic-associated P. Orthophosphate bound by Fe (III) bridges to the humic surface may be released through UV irradiation because Fe (III) can be reduced to Fe (II), followed by the release of both Fe and P, since the stability of the humic Fe (II) complex is relatively low compared to the humic Fe(III) complexes. And humic substances can be degraded by UV irradiation, thereby releasing Fe and P [32]. Phosphatases may release P from P esters bound to the humic surface in solution.

Among the phosphate esters in soil, phosphate monoesters are dominant, and among phosphate monoesters, phytate (myo-inositol-hexakis phosphate) is the dominant phosphate ester in soil [33,34,35,36]. Phosphate monoesters dominate in soil compared to diesters probably because of the stronger bonding of monoesters to the soil solid phase.

The importance of phytate is also underlined by the fact that about 51 million tons of phytate is produced per year in fruits and crop seeds as a P storage compound which is, considering the stored P quantity, about two-third of annual mineral P fertilizer application worldwide [37]. But it should be emphasized that phytate is the most important organic P source in soil if well-defined organic P molecules are considered. Unknown P_o_ often dominates in soils [33,38,39]. Unknown P_o_ is mainly P associated with humic substances of medium to higher molecular masses [16,35]. Higher-phosphorylated inositol phosphates are strongly bound by the soil solid phase. McKercher and Anderson [40] demonstrated that phytate sorption to the soil solid phase is higher than that of the orthophosphate anion. Even inositol trikisphosphate is stronger-sorbed than the orthophosphate anion, whereas inositol monokisphosphate sorption to the soil solid surface is lower than that of the orthophosphate anion [40], calculated on a P basis. Phytate is strongly bound to inorganic surfaces such as Fe/Al oxides or clay minerals [41] and phytate is also strongly bound to humic substances [16,35], resulting in a high proportion of organic P being in higher-molecular-mass fractions in soils where it is firmly held [42,43,44].

A recent paper on soil P_o_ and molecule masses postulates a “molecular size continuum of soil organic phosphorus”, which partly contradicts the work of R.L. Thomas and coworkers from the 1960s and 1970s [45]. The results of Reusser et al. [45] should be treated with caution since the mode of P_o_ extraction and molecule mass determination by size exclusion chromatography, as used by Reusser et al. [45], strongly affects the experimental outcome by disrupting a part of high-molecular-weight P_o_ molecules, leading to the experimental artifact of an organic P molecular continuum.

Figure 3 shows an overview of the P forms in soil, their immobilization reactions and their transfer reactions to the soil solution.

## 3. Plant Strategies to Acquire Phosphate from Soil

Higher plants take up P from soil solution as the orthophosphate anion H_2_PO_4_^−^ [46,47] and possibly the HPO_4_^2−^ anion [48,49]. Organic P esters are assumed not to be taken up by roots [47].

Chen and Barber [50] suggested, by means of modeling P uptake by soil-grown maize roots, that the uptake of the HPO_4_^2−^ anion is lower than that of the H_2_PO_4_^−^ anion by about a factor of 10. The problem with P uptake by soil-grown roots is that the P transfer into the roots increases with a decreasing pH [46,51] and, simultaneously, the P species distribution in the soil solution is affected by changing the pH [46,52]. And to make it more complex, the bulk soil solution composition is of small relevance; instead, the composition at the root cell walls where the transfer of P into the root plasmalemma membrane takes place is relevant. When calculating the P species distribution of the bulk soil solution of the soil solution around the cell walls, striking differences are observed because the pH is lower and the Ca^2+^ activity is higher in the soil solution in close proximity to the cell walls [46]. Also, MgHPO_4_° may be an important P species in the soil solution; other Ca-P and Mg-P species are of minor relevance (p. 195 of Ref. [13]; p. 134 of Ref. [52]). Summarizing the results, at pH values below 6.0, H_2_PO_4_^−^ is the dominant P species in the soil solution. At pH values above 7, CaHPO_4_°, HPO_4_^2−^ and MgHPO_4_° are the most important species, accounting for more of 85% of the species in the soil solution at pH 8.0 (p. 195 of Ref. [13]; p. 134 of Ref. [52]). That means that at a high pH, species such as Ca-P and Mg-P may restrict the uptake of orthophosphate from the soil solution.

The concentration of P in the soil solution is between <1 and about 10 [µmol P/L] [8]. In highly overfertilized soils, P soil solution concentrations are often considerably higher. Phosphate is transported to roots in soil nearly exclusively by diffusion in a soil solution [7,8]. The diffusive P flux to roots strongly depends on the P concentration gradient in the soil solution. If the P soil solution concentration is low, then the maximal concentration gradient dc_l_/dx (with *c_l_* as the soil solution concentration and *x* as the distance from the root surface) is low. Diffusive P flux to the roots is, however, strongly related to the concentration gradient. Because the diffusive flux strongly depends on the concentration gradient, the flux is low if the P soil solution concentrations are low and is, at very low concentrations in the soil solution, negligibly small. Assuming a P soil solution concentration of zero at the root surface, Barraclough [53] calculated the minimal P soil solution concentration to achieve high yields of winter wheat, at14 [µmol P/L], to maintain the required P influx. The increase in P soil solution concentrations is a central step to improve the acquisition of soil phosphate.

The results of Föhse et al. [54] give some insights into the mechanisms by which plants can acquire P from soils under the conditions of restricted P availability. Föhse et al. [54] investigated the acquisition of P from soil by seven plant species in a pot experiment varying the P availability in soil by increasing P applications stepwise. One of the plant species investigated, onion, required a soil solution P concentration of 6.9 [µM] to reach 80% of the maximum yield. Onion has a shallow root system without root hairs, and a relatively low root/shoot ratio [cm root length/mg shoot dry mass]. In contrast, the two graminaceous species in this investigation, wheat and ray grass, had, by far, the highest root/shoot ratio and a P soil solution threshold of 1.2–1.4 [µmol P/L] to achieve 80% of maximum yield. Oil rapeseed, with a similar low P solution threshold of about 1.4 [µmol P/L] had a root/shoot ratio which was 5–10 times lower than that of the graminaceous species, but showed a very high P influx (mol P per time unit and root length) which was higher than that of the graminaceous species by a factor of 1.5 to 10 [54]. Rapeseed and the graminaceous species realized different strategies to acquire P under P-restricted soil conditions. The graminaceous species extended the root surface which supplies the shoots, whereas rapeseed increased the P uptake per root and time unit, possibly by changing the P solubility in the rhizosphere soil, as suggested by Hoffland et al. [55].

Such morphological and physiological adaptions to soil P deficiency, which increase the P uptake from soil, are well known and will be critically discussed in the next chapters.

## 4. Morphological Adaptions of the Roots to Low P Availability in Soils

As P availability decreases in soil, the root/shoot ratio is often increased [54,56,57,58]. Also, the number and length of the root hairs is often increased with decreasing P availability [59,60]. Both adaptions are in the same direction to increase the root surface in relation to the shoot mass. A similar mechanism operates if higher plants form associations with mycorrhizal fungi which may strongly extend the soil volume for P uptake [61]. The extension of the exploited soil volume by the formation of root hairs or mycorrhizal hyphae is a similar mechanism to improve nutrient acquisition, the extension zone often being higher for fungal hyphae, as shown by Jungk [62].

Brown et al. [63] conducted calculations on the role of root hairs in soil P acquisition, including cost–benefit considerations, and concluded that longer and long-living root hairs are a good compromise between effective P acquisition, even in soils low in available P and moderate costs of consuming assimilates in the roots for P acquisition.

But it should be noted that at low P soil solution concentrations, lower than about 2 [µM P], morphological adaptions to increase the nutrient absorbing surface do not adequately improve P uptake by roots because of the severely restricted P diffusion to the roots. The quantitative experimental results and model calculations presented by Claassen ([64], p. 284) show this. Claassen [64] measured and calculated the P uptake by field-grown sugar beet plants using a simulation model without or with the inclusion of root hairs [65]. In the experiment, the P soil solution concentrations varied from 0.9 to 22.5 [µmol P/L] by different rates of P fertilizer application.

In the month with the highest P uptake, July, the actual P uptake by the sugar beet plants was higher than that calculated by a nutrient uptake model at low P soil solution concentrations by a factor of 5–6. The inclusion of root hairs in the calculations only slightly decreased the difference between the calculated and measured P uptake at low P soil solution concentrations, between 0.9 and 2.7 [µmol P/L] ([64] p. 284). In the treatment with the highest rate of P fertilization and a soil solution concentration of 22 [µmol P/L], the difference between the measured and calculated P uptake by the sugar beet plants was relatively low.

The conclusion from the results of Claassen [64] is that most of the P uptake by sugar beet plants at low P soil solution concentrations is not accounted for in the model calculations by including the extent of the root system. And the formation of root hairs in sugar beets only slightly improves the calculated P uptake if the P soil solution concentrations are low. This supports the conclusion that at low P soil solution concentrations, the increase in P-absorbing root surface does not sufficiently improve P uptake.

## 5. Physiological Adaptions of Plants to Low Soil P Availability

Total P in cultivated soils is mostly in the range between 200 and 2000 [mg/kg soil]. Assuming 2 µM P in the soil solution and 20% water content [*w*/*w*] leads to values of less than 0.01% of the total P actually being in the soil solution. To improve P acquisition from soil low in available P, plants have developed strategies to mobilize, i.e., dissolve soil solid P which is not easily released by the desorption from the soil solid phase as a result of root-induced P depletion of the rhizosphere soil solution. The main mechanisms of P mobilization are acidification of the rhizosphere or/and the release of di- and tricarboxylic acid anions. The release of carboxylates, with or without the coupled release of protons, can be an efficient way to dissolve or mobilize sparingly soluble P and increase soil solution P concentrations in the rhizosphere [66,67,68].

## 6. Release of Carboxylates by Roots and Their Impact on P Uptake by Higher Plants

A broad number of plant species increase the release of organic acid anions during P starvation (see for review [47,68,69]). The question is whether P mobilization by root-released carboxylates is a promising way to improve P uptake by plants under conditions of restricted soil P availability. Several cultivated and many wild-plant genotypes may be dependent on the release of carboxylates and the subsequent P mobilization under P-limiting conditions [47,68].

However, the importance of carboxylate release for P mobilization and P uptake by plants has been questioned. Several arguments were made to show that P mobilization by carboxylates is of low importance for P uptake by most cultivated plant species but may be important for wild species, well adapted to low-P soils.

1. The release of carboxylates by roots into the rhizosphere must be very high to reach values above 1 [mmol carboxylate/L] in the soil solution which is required to dissolve soil P [69,70]. Such high values have been shown only for wild plants and few cultivated plants such as white lupin, but may not represent a central mechanism of improving P acquisition from soil by most cultivated plants.

2. The costs of assimilates provided for P mobilization are very high compared to other mechanisms by which P uptake by plants is improved, mainly due to the increase in root hair formation, root hair surface and root/shoot ratio [63].

3. Most aliphatic di-and tricarboxylic acids which are released into the soil solution are easily mineralized in its free dissolved status. Except for extreme conditions, e.g., a carboxylate burst in distinct root regions, the effect of carboxylate release on P solubility in soil may be small because of the low persistence of carboxylates in the soil solution.

In the next chapter, we will consider whether a carboxylate efflux is a central mechanism to improve the acquisition of P from soils under P-limited conditions.

## 7. The Mechanisms by Which Carboxylate Release by Roots Affects Soil P Solubility and P Uptake

Soils low in available P exhibit a low-P soil solution concentration and often high P buffering. They may, however, possess relatively high values of total P (P_t_). Under these conditions, an increase in soil P solubility and an increase in the desorption of soil solid P are central to improve P uptake by roots from soil.

Phosphate desorption from soil is often increased if the soil pH is rather low or high, whereas within a medium pH range, P desorption and P solubility are relatively low [71,72,73].

The effect of carboxylate release by roots on P soil solubility often strongly exceeds the effect of soil pH variations on P solubility [66].

Arguments 1–3 against the central role of carboxylates in P mobilization and P acquisition by the roots of higher plants are partly misleading. The efficiency of carboxylates to mobilize soil P depends on the parameters of the carboxylate molecules. Fox et al. [74] investigated the effect of a variety of mono-, di- and tricarboxylates on P mobilization in a B_h_ horizon of a forest soil, and found that the ability to solubilize soil P was positively related to the stability constant of the respective carboxylate-Al complexes (note that in the soil investigated by Fox et al. [74] oxalate-extractable Al was very high and oxalate-extractable Fe was very low). Citrate and oxalate were the carboxylates with the highest potential of both P and Al mobilization [74]. This is in agreement with the results of Gerke et al. [66,67], who found that citrate was the most effective P-mobilizing agent, followed by oxalate.

The effect of carboxylates on the solubilization of soil P is concentration-dependent. But the most critical concentration parameter is carboxylate concentration at the soil solid-phase surface, not the soil solution carboxylate concentration [66]. In P-fixing soils where P-binding sites are covered to a low extent by P molecules, there is an abundance of free P-binding sites. In this situation, the reaction of carboxylates with the soil solid surface will lead to a coverage of P-binding sites by carboxylate anions. If the resp. carboxylate is effectively competing with the orthophosphate anion, then the carboxylate will effectively bind as the inner sphere complex to the soil solid surface [75,76]. This has important consequences for the mobilization of the P anion from the soil solid phase. Because of the occupation of P-binding sites by carboxylates, the desorption equilibrium between solid P and solution P is, depending on the carboxylate type the extent of surface coverage by the resp. carboxylate and the binding strength of the carboxylate to the soil solid, shifted to the solution P side. It is, therefore, the carboxylate concentration at the soil solid surface which determines the extent of P mobilization. Also, carboxylates bound to the soil solid phase are stabilized against microbial degradation compared to free carboxylates in the soil solution. Boudot [77] showed, for ^14^C-labeled citrate, that the adsorption of the citrate anion to allophane, imogolite or Al hydroxide strongly retarded its microbial decomposition, whereas the microbial degradation of the free citrate anion in solution was rapid. Jones and Ewards [78] showed similar results of citrate stabilization after sorption to Fe hydroxide.

The fast adsorption of carboxylates released from roots into the soil solution may further accelerate the release of carboxylates since the carboxylate concentration in the soil solution is maintained at a low level so that the re-influx of liberated carboxylates into the roots is low.

By the mechanisms of temporary and local carboxylate release by the roots, its adsorption to the soil solid of the rhizosphere and possible triggering of an increased carboxylate net efflux, the accumulation of carboxylates in soil close proximity to the roots, in the rhizosphere may be achieved.

The adsorption of carboxylates to the soil solid phase may be followed by subsequent reactions.

First, the introduction of di- or tribasic carboxylates induces negative charges to the soil solid, especially in the case of citrate, which further reduces the sorption strength of the orthophosphate anion to the soil solid.

Second, a part of the sorbing Fe and Al sites may be dissolved, and the Al- or Fe-carboxylate complexes are transferred to the soil solution. Often, the dissolution of Fe+ Al by carboxylates strongly exceeds that of mobilized P [74,79,80]. These solubilization reactions are pH-dependent since protonation, complexation and redox reactions initiate the dissolution of solid Fe [81].

Third, besides Fe/Al oxides [14] and clay minerals [22] as inorganic binding sites, humic Fe(Al)-P complexes are strongly affected by the release of di- and tricarboxylic anions by roots into the rhizosphere [16,68]. The mechanisms for P desorption from solid-phase humic substances are similar to those from inorganic complexes. And an additional mechanism of P mobilization can be found if P is mobilized by carboxylates from humic Fe (Al)-P complexes. Carboxylates themselves dissolve humic subunits from the soil solid phase, as shown by Gerke [23] and is illustrates in reaction 2 and 3.

The P desorption reaction from humic substances induced by citrate is similar to the P desorption reaction from inorganic surfaces. Also, Al-citrate complexes may be dissolved [23]. And what is different compared to inorganic P sorbing sites is the dissolution of humic subunits with their bound Al (or Fe) and P. By the release of citrate into the rhizosphere, humic substances and their bound Al/Fe and P are released as complexes into the soil solution, which can be measured or determined by P species calculations of the rhizosphere soil solution [52].

Humic metal-P complexes can be the dominant P form in soils [16]. Especially in Podzols and Andosols, these P forms may be abundant [16,82,83] but the humic complexes will also contribute to P forms in other soils. An approach to measure the contribution of humic Fe(Al) surfaces to P sorption in soil is described by Gerke [16]. Van der Zee and van Riemsdijk [84] determined the P sorption capacity of non-calcareous soils by extracting P, Fe and Al with an oxalate solution, as described by Schwertmann [85], the Tamms reagent to be around 0.5 [mol P_ox_/mol Fe_ox_ + Al_ox_] or even higher. This strongly indicates that humic metal complexes account for an important part of P-sorbing sites in non-calcareous soils which were investigated by van der Zee and van Riemsdijk [84]. Humic metal complexes have a sorption capacity (maximum) approaching 1 [mol P/mol Fe + Al], whereas, for example, the sorption capacity of poorly ordered Fe oxide is around 0.1 [mol P/mol Fe] [27]. Oxalate-extractable Al + Fe is assumed to extract the amorphous part of Fe and Al oxides responsible for inorganic P-sorption sites in soil plus a great part of organically complexed Fe and Al [16]. The alkaline pyrophosphate-extractable Fe and Al quantity according to Mc Keague et al. [86] are considered to extract organically complexed Fe and Al, mainly humic Fe(Al) complexes. If the quantity of pyrophosphate-extractable Fe and Al is high compared to oxalate-extractable Fe and Al, then the P sorption to humic Fe(Al) may be important and even dominant in the soil considered [16].

The release of P from humic metal complexes by carboxylates in P-fixing soils is higher than from P bound to inorganic surfaces such as oxides or clay minerals [66,80], partly due to the dissolution of the humic complexes. The diffusion of P into interior areas of the humic aggregates can be assumed as a slow reaction process similar to diffusion into crystalline or amorphous Fe/Al oxides; however, this has not yet been shown in experiments.

The interactions between roots and soil humic substances have to be considered with an even more complicating view. The term “soil root crosstalk” has been introduced, which involves soil humic substances [87,88,89]. The basis for the humic substance-mediated soil–root crosstalk is that soil humic substances affect the morphology and physiology of higher plants partly by acting as plant hormones. The extent of these effects depends on the composition of the soil humic substances, their molecule masses and their solubility. For example, Nardi et al. [90] showed that physiological effects of humic substances on higher plants strongly depend on the apparent molecular masses of the humic substances. The molecular masses and solubility of humic substances, however, are strongly affected by carboxylates, as shown in previous findings [91,92,93,94,95,96]. The excretion of carboxylates induces the reduction in apparent molecular masses of the humic substances, its transfer into the soil solution and the simultaneous liberation of metals such as Fe(III) and Al(III) from the humic phase, and finally the liberation of P.

## 8. Solubilization of Phosphate Esters by Carboxylates

The mobilization of phosphate esters is also affected by carboxylates. Diesters may not be strongly bound to the soil solid phase with the consequence that the biological, biochemical and chemical reactions of this group of P esters will be affected by carboxylates to a smaller extent. A strong accumulation of P diesters in soil is not reported in reviews on soil organic P probably because of the relatively low affinity to the soil solid phase [38,39,97].

The consideration of the mobilization of phosphate esters in soil by carboxylates will be limited here to inositol phosphates as the dominant P-ester forms in soil probably because of the strong affinity of the higher phosphorylated inositol phosphates to the soil solid phase [33,36,40]. Among inositol phosphates, phytate is the dominant species [36]. The reaction of inositol phosphates, and mainly phytate with inorganic surfaces, is formally similar to that of the orthophosphate anion to the same surfaces, both forming inner sphere complexes with the Fe- and Al-OH surface groups of clay minerals and oxides. The extent of reaction is different considering P and phytate concerning the P sorption capacity of the inorganic adsorbents [98]. The phytate molecule is bound to ferrihydrite and hematite by the reaction of two of the six phosphate groups of the phytate to the oxides [99,100]. In the case of goethite, it is assumed that four phosphate groups of the phytate molecule are bound to the oxide surface [99]. Yan et al. [100] showed that a significant desorption of phytate bound to hematite is achieved through the reaction with 0.02 M citrate solution, whereas water or a 0.02 M KCl solution hardly release phytate from hematite at a low phytate coverage (<40%). Amadou et al. [98] compared the uptake of different P sources bound to different minerals, clay minerals, Fe oxide and Al oxide by ryegrass in a pot experiment, and found that phytate was strongly bound by clay minerals and Fe or Al oxides and was mostly prevented from P acquisition in contrast to orthophosphate or other organic P sources. A mobilization step may be required for the acquisition of phytate bound to inorganic sorbents.

Adams and Pate [101] investigated the P acquisition by *white lupin* (*Lupinus albus* L.) and *narrow leaf lupin* (*Lupinus angustifolius* L.) in a quartz sand culture, and in P-fixing soil from orthophosphate (P_i_), glycerol phosphate and phytate. White lupin shows a very efficient way to excrete carboxylates, mainly citrate, in specialized root regions where cluster roots are formed. These cluster roots show a high citrate efflux [102,103,104]. Narrow leaf lupin does not form cluster roots. In quartz sand with limited P fixation, both lupin species took up P from P_i_ and phytate at the same rate. In P-fixing soil, narrow-leaf lupin and white lupin took up P from Pi and glycerol phosphate at similar rates. But the rate of uptake from phytate in soil was almost nil in narrow-leaf lupin, whereas white lupin showed increasing P uptake with increasing phytate application rates. The local high-level release of citrate by white lupin may have increased the solubility of phytate which it could, in turn, be hydrolyzed in the soil solution by phytases (the hypothesis that phytases may hydrolyze phytate bound to the soil solid phase and mobilize P_i_ is discussed in the Section 9.

Similar to the orthophosphate anion, phytate is also bound to humic substances via Fe or Al bridges [35,105]. Inositol phosphates are often found in the high-molecular-weight fraction of soil organic matter [43], and are strongly bound and can be released by the harsh extractant 6 M HCl at 100 °C [44]. A more recent investigation on the molecular size of soil organic phosphorus suggested a continuum in molecular sizes of soil organic phosphorus [45]. However, the authors extracted soil organic P with a mixture of a 0.25 M NaOH and 0.05 M EDTA. NaOH as an extractant has been shown to reduce the molecular weight of soil organic phosphorus [42]. And the effect of the strong complex citrate on the apparent molecular weight of humic-associated P has been discussed before in this paper. The mobilization of soil phytate is often related to its mobilization in the high-molecular-weight fractions of soil organic matter, namely in humic substance fractions. Similar mobilization reactions for humic Fe(Al)-phytate complexes and for humic Fe(Al)-P_i_ complexes may operate after carboxylate release by roots; the desorption of phytate, the release of Al or Fe as a carboxylate complex, and the simultaneous release of phytate and the dissolution of humic Fe(Al)-phytate complexes as a new P species in the soil solution are shown for the orthophosphate anion [23]. However, the confirmation requires further experiments on the acquisition of humic-associated phytate by plants.

Since the phytate molecule is not directly taken up by plant roots, the orthophosphate anion must be released from the phytate molecule by phytases. Phytases are also assumed to mobilize P by hydrolyzing the P anion from phytate adsorbed into the soil solid surface [47,106].

## 9. Phytase Mobilization of Soil P from Phytate

Phosphate ester-hydrolyzing enzymes such as phytases are essential for the acquisition of organic P from soil since the P species which are taken up by the plants are the orthophosphate anion species. The function of phytases in the process of soil P acquisition is still a matter of debate.

Higher phosphorylated inositol phosphates and mainly phytate are quantitatively the most important P_o_ form in soil [36].

At present, the discussion is focused on the question of whether the hydrolysis of phytate or the substrate availability of the phytate is the limiting step of P acquisition from soil phytate [35,36,107].

Based on the results from their study, Jarosch et al. [107] suggested that the enzyme availability or the hydrolysis of phytate is the limiting step assuming that the limited phytate availability (solubility) is not the limiting factor. The authors investigated 10 soils by preparing soil/water suspensions (1:10), and added different enzymes, among them which was phytase. They measured the release of P_i_ after the reaction. Only in one of the ten soils was inorganic P released after the addition of phytase. It may be concluded that enzyme limitation was not responsible for the P acquisition from phytate with the exception of one of ten soils. But even more important, Jarosch et al. [107] added EDTA to the suspensions to “reduce the soil P_i_-sorption capacity and thus to increase the recovery of enzymatically released orthophosphate”. However, EDTA, as a strong Fe(III) and Al(III) complex, indeed affects the P sorption and desorption equilibrium in soils. Similar to citrate, EDTA affects the equilibrium P_i soil solution_ vs. P_i solid phase_ by increasing the solution side so that P_i_ is desorbed. Also, phytate is bound to the soil solid phase to the same Al(Fe)-OH groups which bind P_i_ so that the EDTA addition will also affect the equilibrium phytate _soil solution_ vs. phytate _soil solid_, and shift it to the solution side. It is inacceptable to assume that EDTA only prevents the re-adsorption of newly formed orthophosphate, ignoring the effect of EDTA on the equilibrium of different P species between solid and solution. Based on a misleading experimental setup, Jarosch et al. [107] concluded that the low phytase activity is the rate-limiting step of P acquisition from phytate.

There are contrasting results on the acquisition of P from phosphate esters, mainly from phytate as the dominant form. Gerke [35] summarized these results. Tarafdar and Claassen [108], Beißner [109], and Lung and Lim [110] found that the restricted phytase activity is not the rate-limiting step in P acquisition from phytate. Beißner [109], and Lung and Lim [110] instead showed that the availability of phytate, its strong sorption to the soil solid phase, is the limiting step in P acquisition from phytate. The accumulation of phytate and higher phosphorylated inositol phosphates in soil as the main definite organic P forms may be due to the strong bonding to the soil solid phase, which, in turn, may stabilize it even against hydrolysis and microbial degradation (for a recent balanced review, see Liu et al. [36]). The conditions under which phytases may improve the P acquisition from phytate sorbed and immobilized as reported by Hayes et al. [111] or George et al. [112]. In both investigations, the role of phytases was evaluated in sterile agar, and it was assumed that no phytate sorption occurs. But the Ca^2+^ activity was relatively high at 4.0 [mM]. Note that the Ca^2+^ activity was about a factor of 10 lower in the experiments of Tarafdar and Claassen [108], Beißner, [109], and Lung and Lim [110], which may have reduced the probability of Ca-phytate precipitates, allowing for a true differentiation of the effect of the hydrolysis and phytate sorption. Ca-phytate precipitates may have reduced phytate solubility even in non-sorbing agar. If, however, the phytate solubility remained relatively high even with a proportion precipitated with Ca^2+^, then soluble phytate may have been hydrolyzed with a higher rate after the addition of phytases, inducing further dissolution of the Ca-phytate precipitates. The chemical equilibria between solid phytate and solution phytate in soils are clearly different from agar substrate. In P-fixing soils, the soil solid phase is a sorbent for both orthophosphate and phytate, the phytate retention being probably more pronounced. In a rare study of IHP soil solution concentration measurements, Espinoza et al. [113] found, in a soil leachate, that IHP accounted for about 3% of the P in solution, whereas orthophosphate accounted for 67%. Based on P the quantity of IHP sorbed into the soil solid phase, its sorption is often higher than that of the P_i_ anion at a definite P equilibrium concentration [35,98,99,100]. That means that in soils, the solution concentration of IHP is expected to be very low and may be different from the chemical equilibria in agar systems with Ca-phytate as fresh precipitates. The increased hydrolysis of phytate in agar after the addition of phytases may induce further dissolution of phytate in this substrate with a relatively high phytate solubility, whereas in soil, the hydrolysis of soil solution phytate may have a negligible effect on the desorption of phytate.

Another hypothesis related to the role of phytases in the dissolution of solid-phase phytate P has recently gained some attention. Phytases may dissolve phytate P bound to the soil solid phase by hydrolyzing phytate at the soil solid phase. This reaction is considered the main mechanism which makes phytate P plant available [47]. The simultaneous release of carboxylates by the roots is assumed to limit the re-adsorption of the released P_i_ anion to the soil solid phase.

This P_i_ liberation mechanism has been postulated by referring to the scientific source by Garcia-Lopez et al. [106]. The authors conducted an experiment with cucumber cultivated on ferrihydrite-coated sand. Phytate was the single P form which was applied. The uptake of P by the cucumber plants decreased with an increasing concentration of ferrihydrite in the substrate. The authors also measured that most of the P in the experiment with plants was recovered as the orthophosphate anion. They concluded that the phytate-P was mostly hydrolyzed. They also concluded from their results that the sorption of phytate to the ferrihydrite surface was not the limiting step of phytate P availability but the re-adsorption of the liberated orthophosphate to ferrihydrite [106]. However, their conclusions drawn from the experiment should be considered with caution. In this experiment, the P loading of ferrihydrite with phytate was very high, almost reaching P saturation in the lower ferrihydrite treatment with 0.5 [mol P/mol Fe_ox_ + Al_ox_] (see [84]). This means that in the experiment of Garcia-Lopez et al. [106], the initial concentration of phytate in solution was probably rather high, although the authors did not measure this important parameter in their experiment. If this was the case, then the hydrolysis of phytate in a solution may induce further phytate desorption from the solid phase from ferrihydrite. Similar to the conditions of the experiments of Hayes et al. [111] and George et al. [112], the phytate solution concentrations may be rather high and may not represent the chemical equilibria in most agricultural soils with very low soil solution phytate concentrations. The experiment discussed here is no proof for the hypotheses that soil solid-phase phytate can be directly hydrolyzed by phytases.

A second aspect makes the interpretations by Garcia-Lopez et al. [106] even more questionable. An examination of plant parameters in the paper of Garcia-Lopez et al. (Table 1 in Ref. [106]) shows that P deficiency may not be the yield-limiting step in their experiment. The P concentrations in the shoots, even with the highest ferrihydrite treatment, were higher than in the shoots with non-sorbing quartz sand treatment, indicating that reduced plant yields in the treatments with ferrihydrite may not be caused by P deficiency. And similar to Jarosch et al. [107] who used EDTA as a complexant, Garcia-Lopez et al. [106] attributed the role of complexants, here carboxylic anions, to avoid exclusively re-adsorption of the liberated orthophosphate anion, ignoring the role of carboxylates for the even-interacting equilibria between phytate s_olution_/phytate_soil solid_ and P_i solution_/P_i soil solid_. The addition of a complexant competing with phytate and orthophosphate at the soil solid affects the quantitative distribution of all four interacting P species.

There is no proof that phytases mobilize phytate from the soil solid phase in P-fixing soils. The role of phytases probably lies in the hydrolysis of soil solution phytate after its mobilization, for example, by carboxylates.

## 10. The Relevance of Phosphate-Mobilizing Microorganisms for Phosphate Uptake by Higher Plants in P-Deficient Soils

Barrow and Lambers [73] showed ways of experimental approaches and ways of calculations to answer this question, and they also showed the possible ways of misinterpretations which may overestimate the contribution of P solubilization by microorganisms to P uptake by higher plants.

If microorganisms selected for the P solubilizing effect decrease the pH of the soil solution, then the effect of the microorganisms on P uptake may be misinterpreted since the decrease in soil pH in the range of 5.0–7.0 increases P uptake from soil without changing soil P solubility [73]. To differentiate the effect of microorganisms on soil pH and P soil solubility, Barrow and Lambers [73] used a modified Mitscherlich equation.

To apply this equation, plant experiments with increasing P applications, and with and without the inoculation of P-mobilizing microorganisms are required. If the P solubilizing effect by the added microorganisms increases the plant yield, then the additional P release, Δ P, by the microorganisms will shift the yield response curve (yield vs. P applied) to the left by Δ P [73]. If the effect of inoculation on P uptake is a pH effect, then the slope of the response curve will be higher compared to the response curve without inoculation [73]. The authors found that in most studies using inoculation with P-solubilizing microorganisms, there was only one P level, so that the solubilizing effect and the pH effect cannot be distinguished [73]. Few exceptions exist. The field results by Whitelaw et al. [114] and the pot experiments by Gomez-Munoz et al. [115] showed a higher slope of the P response curve, indicating a pH effect of the inoculation. Schütz et al. [116] conducted a meta-analysis on the inoculation with P-solubilizing microorganisms, and found that the response to inoculation was small at low soil P levels and that a response was more likely at a higher pH, between 7.3 and 8.3. Barrow and Lambers [73] concluded that soil acidification by the inoculum is the main effect on plant P uptake, consistent with the results of the meta-analysis by Schütz et al. [116].

The excretion of mobilizing agents such as citrate and oxalate represent a mechanism of plants to acquire more soluble P. If the roots deliver plant assimilates to P-solubilizing microorganisms which aim to produce mobilizing agents, additional assimilates for the maintenance of the microorganisms are required. It will probably cost more energy compared to direct release of carboxylates by roots.

Widely occurring P mobilization in soil by solubilizing microorganisms is yet to be proven considering the approach by Barrow and Lambers [73]. This does not exclude an important role of P mobilizing microorganisms in soil.

## 11. Heterogeneity in the Rhizosphere: A Clue for the Effect of Root Carboxylates on P Uptake by Plants

Spatial and temporal variations in plant and soil parameters probably play an important role for the quantitative effect of carboxylate release by roots on soil P mobilization, P acquisition and P uptake by plants. Carboxylates are excreted in a distinct region behind the root tips in rapeseed [117] in specialized cluster roots, e.g., white lupin or yellow lupin of a definite age of the roots [102,103,118] and also in many members of Proteaceae and some Cyperaceae [119,120,121]. Carboxylate release by roots is also important in red clover and alfalfa, and, to some extent, white clover [66,67]. An increased release of oxalate following P starvation was shown for sugar beet and spinach [66,122]. The carboxylate release is non-uniform along the root axis, often with a maximum release about 1.5 cm behind the root tips [57,109,122,123]. Also, the plant uptake parameters according to Michaelis–Menten kinetics will vary depending on the age of the roots and the formation of root hairs, which itself depends on the age of the roots, as summarized by Jungk [62], and the nutrient status of the plants as shown by Jungk and coworkers [124]. And temporal and spatial varying plant and soil parameters are also interacting. Lambers [47] stated that in strongly P-deficient soils, the physiological adaptions of the uptake parameters, e.g., according to the Michaelis–Menten kinetics equation, will not significantly contribute to an increased P uptake. If the P soil solution concentration is low then the diffusive flux to the roots will be low even if I_max_ and k_m_ are strongly changed as seen from model calculations ([64], p. 157). But if in a P-solubilizing cluster root region the P soil solution concentration is increased, then the spatial and temporal variation of P uptake parameters of the roots may be important for increased P uptake and influx. Vorster and Jooste [125] found that the cluster roots of Proteaceae plants showed increased I_max_ and lowered k_m_ values compared to ordinary roots, consistent with a higher metabolic uptake of P in cluster root regions compared to ordinary roots.

Another relevant question with respect to P mobilization by carboxylates from the soil solid phase is the question on time or more precisely, should the release of soil solid P be considered in terms of equilibrium between the solid and the solution phase, or are the temporal variations, the kinetics of P liberation and uptake, as affected by carboxylates, more important. Most studies dealing with P availability, P acquisition and P uptake by plants affected by carboxylates are assuming explicitly or implicitly an equilibrium relationship between soil solid P and solution P.

Rare exceptions are two studies which considered the kinetics of the dissolution of solid soil P as affected by carboxylates. Fox et al. [79] investigated the kinetics of P desorption from A_h_, B_h_ and B_t_ horizons of a sandy soil by formate and oxalate using a batch technique. Formate had no effect on the P solubility, whereas oxalate mobilized inorganic and organic P in the B_t_ and B_h_ horizons. Gerke [80] used a continuous flow technique to evaluate the effect of different rates of citrate on the P solubility in a humic podzol and luvisol. This technique has the advantage that the desorbed P is removed, which is more similar to the situation in the rhizosphere where the root is the P sink. The rate of citrate application (0, 10, 20, 50 µmol/g soil) and the mode of citrate application, i.e., single addition before the beginning of the experiment vs. sequential citrate application during the desorption, were investigated, and P_t_ was determined in the eluates. The cumulative P desorption increased with the citrate application rate and was, by far, higher in the humic podzol compared to the luvisol. In the humic podzol, humic metal-P complexes account for about 80% of the total P [83], whereas in the luvisol, P adsorbed into the inorganic surface was dominant. Humic-associated P is relatively easily mobilized by carboxylates compared to P sorbed into clay minerals or Fe/Al oxides. The effect of the mode of citrate application—single addition vs. sequential application—on the kinetics of P desorption depended on the soil and level of citrate application. In general, the P soil solution concentrations were increased by the application of citrate during the first 36 h of desorption from <10 µM P to more than 100 µM P, and then decreased to <10 µM P again from 63 to 72 h [80]. This enormous temporal increase and decrease in P solution concentrations may also be operating in rhizosphere soil after a temporally limited release of carboxylates by the roots.

Without a doubt, strong temporal and local variations in nutrient availability and acquisition may complicate the mathematical modeling of root-released carboxylate on P uptake by plants. Robinson [126] stated that the most influential model to describe water and nutrient uptake by single roots and whole-root systems was going back to Gardner [127]. It was further developed by Nye and coworkers (see [128]), and summarized by Nye and Tinker [129] and Tinker and Nye [7]. The results of this model may show good agreement between the model and experiment [130], or no good agreement [131]. Besides soil heterogeneities, multiple local response systems in plants, e.g., reacting to nutrient distribution inhomogeneities, may not be a simple summation of molecular processes in whole plants [126].

This makes it difficult to predict the uptake of P by plant roots as a result of P mobilization by carboxylates. Simplified models of P mobilization in soil and the uptake of mobilized P by roots may, nevertheless, give a quantitative insight. The bases for such calculations were laid by P.H. Nye [132,133]. Kirk et al. [134] measured the P uptake by rice plants and showed that the plants’ P uptake could be explained by the mobilization of P through the release of protons and citrate. Gerke et al. [67] calculated the uptake of P mobilized by citrate or oxalate, and showed that the P mobilization reactions strongly contribute to P uptake in red clover (citrate) and sugar beet (oxalate).

## 12. The Agronomic Effect: Improving P Acquisition in Crop Rotations, Permanent Grassland and Cover Crops with Carboxylate-Releasing Plants

Several cultivated dicotyledeneous plant species possess the ability to mobilize soil P by the excretion of carboxylates, citrate and oxalate, being the most effective carboxylates with respect to soil P mobilization. P availability may be strongly locally and temporarily improved in low-P soils by the release of carboxylates. Many of the plant species which release carboxylates during P starvation are legumes, grain legumes such as white and yellow lupin [118], or fodder legumes such as alfalfa [52], red clover or white clover [67]. White lupin has not only been shown to improve P uptake from P-deficient soil itself, but it also improves the P uptake by wheat in a mixed cropping system [135,136] and also that of wheat grown after white lupin [137].

Cu et al. [138] also investigated the mixed cultures of wheat and white lupin. Shoot growth and P uptake in wheat were increased by 33 and 45%, respectively, when wheat was grown in a mixed culture with white lupin, whereas the shoot growth and P uptake of white lupin was not affected by wheat as a partner in mixtures. Cu et al. [138] showed that white lupin exploited the soil pool of citrate extractable P, whereas wheat exploited the pool of water-extractable P.

Lelei and Onwonga [139] showed, in a rotation with white lupin and maize fertilized with water soluble-P or rock phosphate P, that white lupin made rock phosphate P available to maize in rates that were found in the water-soluble P treatment without white lupin in the rotation. White lupin improves the P uptake of plants grown in mixtures with white lupin grown after white lupin as cover crop or following white lupin as the main crop.

Similarly, *yellow lupin* (*Lupinus luteus* L.) also forms cluster roots which show a maximum citrate release rate which is about 70–80% that of white lupin [118]. Yellow Lupin has been used to improve nutrient-impoverished acid sandy soils in Germany in the 19th century. It was a now-famous farmer, Albert Schultz-Lupitz, who managed a farm in the “Altmark”, today in Sachsen-Anhalt, Germany, and grew yellow lupin to feed sheep and improve the sandy soils as a cover crop. Four periods of managing by Schultz-Lupitz are distinguished: a. the *yellow lupin* period, b. the liming period, c. the kainit period and d. the phosphate period. The first period gave positive results probably due to the N_2_ fixation and P mobilization by the cluster roots of yellow lupin. The liming period (Mergel Periode) was, after some initial success, harmful and caused iron chlorosis in yellow lupin. The third period initiated greater agronomic success since strong K deficiency, and also Mg and S deficiency were ameliorated by kainit [140]. The cultivated area of yellow lupin was greatly extended on northern German acid and impoverished sandy soils, and decreased again after the introduction of mineral P fertilizers to these soils at the end of the 19th century. The case of yellow lupin cultivation in Germany showed the widespread cultivation of a P-mobilizing plant species to make soil P accessible to the plant roots before the introduction of mineral P fertilizers began.

Nuruzzaman et al. [141] investigated the effect of white lupin, faba bean and field pea on the yield of subsequent wheat and found that P mobilizing species increased wheat yield the effect being greater if P fertilizers were also applied.

A pronounced effect of P mobilization on the P uptake may be expected in *alfalfa/clover/grass* mixtures. Alfalfa, red clover and, to some extent, *white clover* release citrate during P deficiency [52,67]. The quantitative determination of the P mobilization and the uptake of mobilized P is difficult for several reasons. *Alfalfa* and *clover* species are N_2_-fixing species where N_2_ fixation itself depends on the P status of the legumes. If P is mobilized by the excretion of carboxylates by alfalfa or clover roots, then about 90% of the mobilized P will diffuse away from the single root which released the carboxylates [133]. Considering that the rooting density of the graminaceous species in the mixture is higher than the rooting density of the legumes by a factor of 5–10, then grass roots will mostly benefit from the mobilization of soil P by legume-released carboxylates in the legume/grass mixtures.

Fertilizer P is frequently reported to promote clover and alfalfa yield in permanent or semi-permanent legume–grass mixtures [142]. This statement is supported by Bi et al. [143], who showed for legume–grass mixtures a strong legume-supporting effect of P fertilization under the conditions of limited rainfall. This effect may be due to the fact that grass species benefit more from the P mobilization by alfalfa or clover species compared to the legumes. Nevertheless, the release of carboxylates by the legumes in mixtures with grasses may be important for the P uptake of legume–grass mixtures.

*Alfalfa/clover/grass* mixtures are an essential part of rotations in organic farming. This is so because nitrogen fixation by the legume part of the mixtures is essential to provide sufficient nitrogen to the crop rotations and permanent grassland. Also, P mobilization and P uptake processes in legume–grass mixtures for fodder production should be considered in more detail for sustainable agriculture.

## 13. Summary and Conclusions

Phosphorus is an essential plant macronutrient with strongly limited P reserves worldwide.

Among the macronutrients, phosphate has, by far, the lowest solubility in soil.

Most of soil P consists of P sorbed to oxides and clay minerals or to humic Al (Fe) complexes, with a continuum of binding strength of P to the soil solid phase, as described and quantified by N. J. Barrow and coworkers.

Among the various compounds of organic P esters, higher phosphorylated inositol phosphates account for an important, sometimes dominant, part of soil P. The inositol phosphates are often bound to humic Al(Fe) complexes, resulting in a medium-to-high-molecular-weight fraction of soil organic P.

Soil P deficiency is a problem worldwide for food production. In P-deficient soils, the soil solution concentrations are low, mostly below 2 [µmol P/L], so that the diffusive P flux to the plant roots is severely restricted. However, the pool of total P may be high even in P-deficient soils.

Under conditions of medium-to-high total P concentration in soil and low P soil solution concentrations, the mobilization of solid-phase P by carboxylates released by plant roots represents a promising way to improve P uptake without or with relatively low P fertilizer applications.

Among the carboxylates, mainly citrate, and to some extent oxalate, are able to mobilize and dissolve soil solid-phase P.

Besides the reactions of soil-released carboxylates with inorganic surfaces such as Fe(Al) oxides or clay minerals, P mobilization by carboxylates from humic metal complexes may be important, since P is desorbed from the humic phase and humic metal-P complexes themselves are dissolved by carboxylates.

The agronomic relevance of root-released carboxylates on P uptake in crop rotations and permanent grassland may be high but has to be further supported by experiments on the laboratory and field scales. Promising results were reported for the cluster-root-forming plant species such as white lupin and yellow lupin in crop rotations. Also, P mobilization in legume–grass mixtures where the legume part, alfalfa and several clover species, release carboxylate which may mobilize P and improve the uptake of P by both the legumes and the grasses, which may be an important contribution to P nutrition in crop rotations, including alfalfa/clover/grass mixtures or permanent grassland. However, the uptake of P mobilized by carboxylates by such mixtures is still, at best, partly understood.

Root-released carboxylates may be essential under low-P soil conditions to acquire soil P by plant roots. Humic-associated P is a soil P form which is more easily dissolvable by carboxylates such as citrate and oxalate compared to P sorbed into oxides and clay minerals.

## Figures and Tables

**Figure 1 plants-13-03478-f001:**
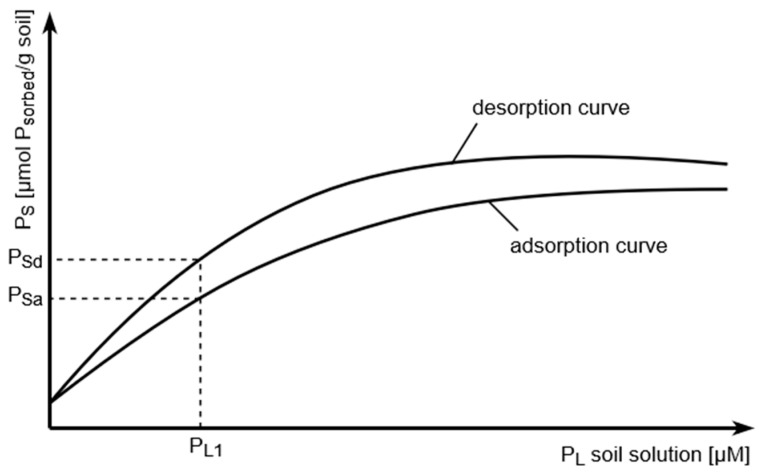
Adsorption–desorption hysteresis of soil phosphate. Note that to achieve a definite P solution concentration, the P coverage of the soil solid phase must be much higher during desorption (Psd) compared to adsorption (Psa).

**Figure 2 plants-13-03478-f002:**
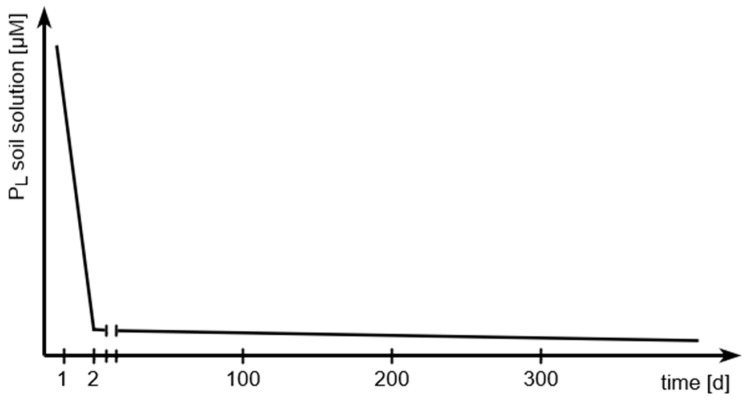
Principal slow reaction between added phosphate and soil.

**Figure 3 plants-13-03478-f003:**
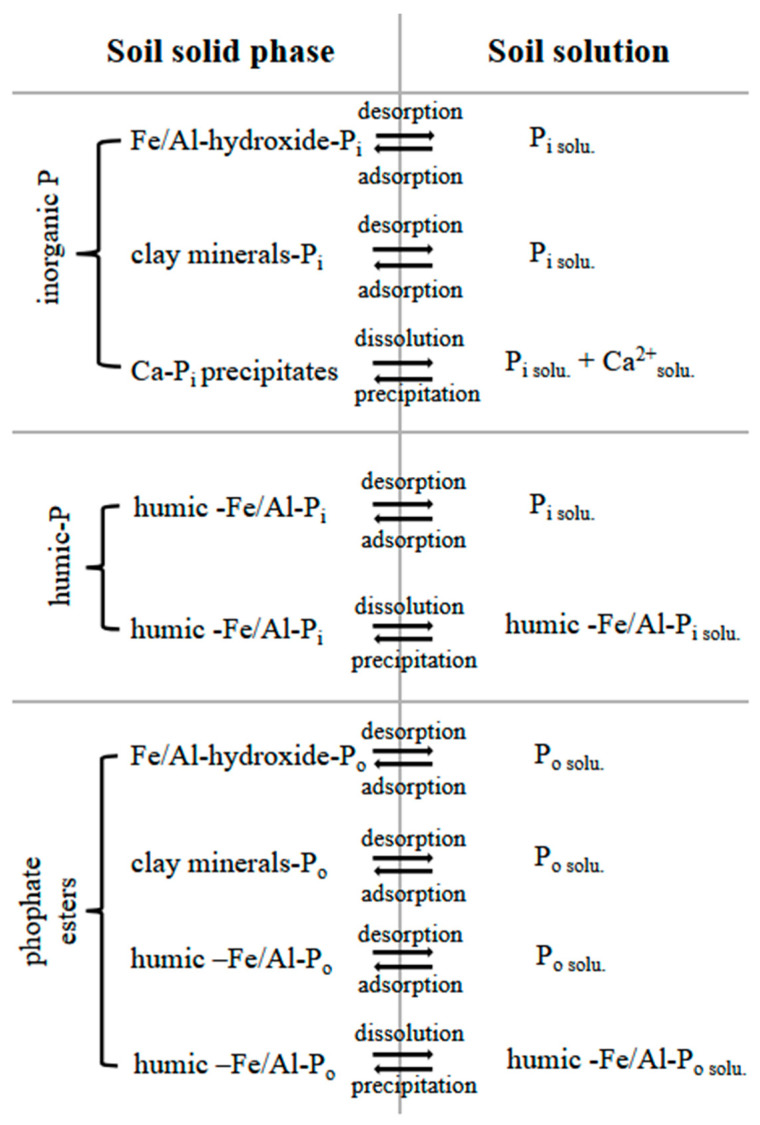
Phosphate at the soil solid phase, P transfer into the soil solution and immobilization reactions affecting soil P availability. P_i_—orthophosphate; P_o_—phosphate ester, solu.—species in solution.

## Data Availability

Not applicable.

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
