# Peer review of "Improving Phosphate Acquisition from Soil via Higher Plants While Approaching Peak Phosphorus Worldwide: A Critical Review of Current Concepts and Misconceptionsâ€"

_plants, 2024, doi:10.3390/plants13243478_

Round 1
Reviewer 1 Report
Comments and Suggestions for Authors
Dear Mr Gerke,
I read your article with great pleasure; the article is very well done! Congratulations on this achievement!! I was particularly interested in the part about phytate.
You made a mistake in line 136. Fe- and Al-P are extracted with NaOH during P fractionation and the Ca-P are extracted with an acid.
You could have gone into more detail about mycorrhiza.
Author Response
Dear reviewer,
thank you for your comments and your statement that you read the article with great pleasure.
Your comment on line 136 points out to an expression of mine which can be misunderstood. I wrote that discrete fractions are associated with Al- or Fe- phosphate under acidic conditions and Ca-phosphates under basic conditions.
The text has been changed to make clear that Ca-P is extracted with acid solutions and Fe/Al-associated P is extracted with NaOH in the fractionation procedures mentioned before.
Regarding mycorrhiza: If the soil solution P concentration is very low then even the formation of mycorrhiza may not improve P acquisition from strongly P fixing soil similar to the intensive formation of long root hairs. In the 1990ies the group of the late Horst Marschner was looking whether mycorrhiza hypae may release P mobilizing agents into soil. To my knowledge these efforts were not successfull and had no successor of research. Therefore I only briefly mentioned this aspect.
Reviewer 2 Report
Comments and Suggestions for Authors
The review submitted to Plants by Jorg Gerke is devoted to how higher plants can obtain phosphate from the soil.
Figures 1 and 2 are unfairly large, perhaps they should be combined to save space. In their current form, they do not convey as much information as they take in the manuscript.
Figure 3 is very poor quality and looks like a table screenshot. It should be redesigned.
The same complaints can be made about Figure 4 - it has no illustrative material, just words and ugly arrows.
Perhaps the author should try to formulate a hypothesis that he has tested during the literature review and answer in the last section whether it has been proven.
It is unfair that the manuscript reviews the phosphate-solubilizing soil bacteria (section 10) in too short words. Although they are discussed in detail in various reviews, including those published in Plants, it seems unfair not to give them the attention they deserve in this manuscript.
Author Response
Dear reviewer 2
thank you for your comments.
- Figure 1 and 2 were strongly reduced in size. I preferred to leave 1 and 2 as seperate figures.
- I think figure 3 is neccessary and that the graphics is acceptable. Fig. 3 brings together all P adsorbents in soil which may be rather new.
- Figure 4 was removed. I also was not satisfied with it but the mechanisms which should be illustrated can not be found elsewhere. Instead of fig. 4 I made reference to one of my pubications (23) which illustrates some of the content I wanted to illustrate with fig. 4.
- The title points out to give an overview on present concepts and misconceptions. And I think that the main hypothesis of the whole ms article is that in strongly P deficient soils P mobilization by carboxylates is essential for P acquisition and uptake by higher plants.
- Concerning the role of P-solubilizing microorganisms I think that the discussion of the Barrow-Lambers paper from 2022 shows that proven effects are rare. This may be a chance for experiments which take into account the comments of Barrow-Lambers mainly the establishment of several P levels. As I noted at the end of the chapter such effects are yet not measured or proven. Considering the comments by Barrow and Lambers this statement may be justified.
Round 2
Reviewer 2 Report
Comments and Suggestions for Authors
I think figure 3 is neccessary and that the graphics is acceptable. Fig. 3 brings together all P adsorbents in soil which may be rather new.
The Fig. 3 look unacceptable in XXI century. All the information presented in this ugly picture will be more useful, if presented in text
The title points out to give an overview on present concepts and misconceptions. And I think that the main hypothesis of the whole ms article is that in strongly P deficient soils P mobilization by carboxylates is essential for P acquisition and uptake by higher plants.
Please, add the hypothesis and whether is was proven to the manuscript.
Concerning the role of P-solubilizing microorganisms I think that the discussion of the Barrow-Lambers paper from 2022 shows that proven effects are rare. This may be a chance for experiments which take into account the comments of Barrow-Lambers mainly the establishment of several P levels. As I noted at the end of the chapter such effects are yet not measured or proven. Considering the comments by Barrow and Lambers this statement may be justified.
Many articles and patents, as well as commercial preparations of solubilizing bacteria, indicate that they 1) have a beneficial effect on linear growth and yield; and 2) allow a reduction in the amount of mineral fertilizer applied while maintaining yield. This should be mentioned in the text, along with the fact that these processes are difficult to measure.
Author Response
Dear reviewer
- I removed figure 3 and replaced it by a probably more viewer friendly figure 3, new.
- I added the hypothesis on the role of root released carboxylates on P acquisition at the end of the introduction and made an additional statement in summary and conclusions on the role of carboxylates on soil P acquisition.
- I removed the term "not measured" in the chapter on the results of Barrow and Lambers (2022). In my opinion the critics made by Barrow and Lambers (2022) are significant and should be meet by future experiments as suggested by these authors. I do not agree with reviewer 2 on this topic however accept that my original term "not measured" may be not adequate.
Round 3
Reviewer 2 Report
Comments and Suggestions for Authors
In the reviewer's opinion, the article has not been improved to a state suitable for publication in Plants. All figures are of disappointingly low quality, the text is not formatted according to the journal template, and some critical points are not reflected in the text.
I believe that the reviewers' comments are sufficient for the Editor to make a fair choice and decision about the advisability of publishing this text in Plants.
Sincerely,
Author Response
The response to reviewer 3 will be given in detail within the letter to the editor.